# Expression of the Endoplasmic Reticulum Stress Marker GRP78 in the Normal Retina and Retinal Degeneration Induced by Blue LED Stimuli in Mice

**DOI:** 10.3390/cells10050995

**Published:** 2021-04-23

**Authors:** Yong Soo Park, Hong-Lim Kim, Seung Hee Lee, Yan Zhang, In-Beom Kim

**Affiliations:** 1Department of Anatomy, College of Medicine, The Catholic University of Korea, 222 Banpo-daero, Seocho-gu, Seoul 06591, Korea; yongsoopark88@gmail.com (Y.S.P.); seunghui6310@daum.net (S.H.L.); 00zhangyan00@naver.com (Y.Z.); 2Catholic Neuroscience Institute, College of Medicine, The Catholic University of Korea, 222 Banpo-daero, Seocho-gu, Seoul 06591, Korea; 3Integrative Research Support Center, College of Medicine, The Catholic University of Korea, 222 Banpo-daero, Seocho-gu, Seoul 06591, Korea; wgwkim@catholic.ac.kr; 4Department of Biomedicine and Health Sciences, Graduate School, The Catholic University of Korea, 222 Banpo-daero, Seocho-gu, Seoul 06591, Korea; 5Catholic Institute for Applied Anatomy, College of Medicine, The Catholic University of Korea, 222 Banpo-daero, Seocho-gu, Seoul 06591, Korea

**Keywords:** retinal degeneration, endoplasmic reticulum, stress response, unfolded protein response, GRP78, retinal glial cell

## Abstract

Retinal degeneration is a leading cause of blindness. The unfolded protein response (UPR) is an endoplasmic reticulum (ER) stress response that affects cell survival and death and GRP78 forms a representative protective response. We aimed to determine the exact localization of GRP78 in an animal model of light-induced retinal degeneration. Dark-adapted mice were exposed to blue light-emitting diodes and retinas were obtained at 24 h and 72 h after exposure. In the normal retina, we found that GRP78 was rarely detected in the photoreceptor cells while it was expressed in the perinuclear space of the cell bodies in the inner nuclear and ganglion cell layers. After injury, the expression of GRP78 in the outer nuclear and inner plexiform layers increased in a time-dependent manner. However, an increased GRP78 expression was not observed in damaged photoreceptor cells in the outer nuclear layer. GRP78 was located in the perinuclear space and ER lumen of glial cells and the ER developed in glial cells during retinal degeneration. These findings suggest that GRP78 and the ER response are important for glial cell activation in the retina during photoreceptor degeneration.

## 1. Introduction

Retinal degeneration (RD) is a leading cause of blindness and is characterized by the irreversible and progressive loss of photoreceptor cells. Age-related macular degeneration (AMD) is the most common RD with a multifactorial cause of progression. Various factors including a genetic contribution, light-induced stress, oxidative stress and inflammation affect photoreceptor cell death during AMD [1,2,3]. For the treatment of wet AMD, anti-vascular endothelial growth factor (VEGF) therapy is a unique, effective treatment option. However, long-term intravitreal injections of an anti-VEGF agent is needed, which can cause complications including ocular hypertension, inflammation, retinal detachment and hemorrhage [4,5]. Moreover, a therapeutic strategy for dry AMD has not yet been established [6]. Thus, studies on the pathogenesis of RD including AMD are needed.

The endoplasmic reticulum (ER) stress response is an important intracellular mechanism of neuronal cell death [7,8]. Stress conditions in the ER can trigger the unfolded protein response (UPR), which increases the number of endoplasmic chaperones to clear the accumulated unfolded proteins and to maintain homeostasis [7,9]. However, when ER stress exceeds the UPR, ER stress undergoes a cell death pathway via the activation of the C/EBP homologous protein (CHOP) [10]. There is increasing evidence that ER stress and ER responses are involved in the pathogenesis of AMD and its progression [11,12]. Previous animal studies revealed that ER stress markers were increased in the retina during photoreceptor degeneration [13,14]. In addition, a drug-induced ER stress model could promote photoreceptor loss in mice [15]. However, there is a contrary report that ER stress could protect photoreceptors in Drosophila [16]. Thus, the relationship between photoreceptor cell death and the ER stress of photoreceptors remains unclear.

Immunoglobulin heavy chain-binding protein (BiP), also known as 78-kDa glucose-regulated protein (GRP78), is a representative ER stress marker that belongs to the ER chaperone. GRP78 regulates protein folding to prevent the accumulation of misfolded proteins and maintains ER homeostasis and cell protection [17,18]. Previous studies have shown increased GRP78 expression in RD models induced by light injury [13,19] and retinal detachment [13,19] and in inner retinal degeneration induced by N-methyl-D-aspartate (NMDA) toxicity [20,21], suggesting the involvement of GRP78 in the pathogenesis of various types of retinal degeneration. However, its localization in the normal retina remains unclear. A few investigators have reported that GRP78 is expressed in the inner nuclear layer (INL) and ganglion cell layer (GCL) in the normal retina [20,21] while others have not detected GRP78 expression in cells in the outer nuclear layer (ONL), INL and GCL in a normal retina [19]. Therefore, detailed information on GRP78 expression at cellular and subcellular levels in normal and RD retinas needs to be elucidated to understand the role of ER stress in the pathogenesis of RD.

We aimed to determine the cellular and subcellular localization of GRP78 in normal retinas and to examine the changes in the GRP78 expression profile in RD induced by light-emitting diodes (LED) using immunohistochemistry and an advanced immuno-electron microscopic technique, correlative light and electron microscopy (CLEM).

## 2. Materials and Methods

### 2.1. Animals

A total of 30 male BALB/c mice at seven weeks of age were used in this study. The mice were kept in a climate-controlled condition with a 12 h light and dark cycle and divided into three groups: normal, 24 h and 72 h after LED exposure (*n* = 10 for each group). All procedures followed the regulations established by the Institutional Animal Care and Use Committee of the School of Medicine, The Catholic University of Korea (Approval number: CUMS-2017-0241-03), which acquired AAALAC International full accreditation in 2018.

### 2.2. Exposure to a Blue LED

The blue LED-induced RD model was described in detail in our previous studies [22,23,24]. Mice were kept in a dark room for 24 h before LED exposure and their pupils were dilated with Mydrin P (Santen Pharmaceutical Co., Osaka, Japan) under dim red-light conditions 30 min before LED exposure. Afterwards, mice were exposed to a 2000-lux blue LED for 2 h. After LED exposure, the mice were kept in a dark room for 1 h and then the mice were moved to a climate-controlled condition with a 12 h light and dark cycle.

### 2.3. Tissue Preparation

At 24 h and 72 h after LED exposure, the mice were anesthetized by an intraperitoneal injection of zolazepam (20 mg/kg) and xylazine (7.5 mg/kg) for tissue preparation. The anterior portion of the eye was dissected and eye cups were fixed in 4% paraformaldehyde for 2 h. After fixation, the eye cups were washed with a phosphate buffer (PB, 0.1 M) and then transferred to 30% sucrose in PB 0.1 M for one night. The tissues were embedded in an OCT compound to prepare the frozen tissue. Sections of the eye cups at the center of the retina of 8 μm thickness were obtained.

### 2.4. Terminal Deoxynucleotidyl Transferase dUTP Nick End Labeling (TUNEL) Assay

After washing, the retinal sections were incubated in a permeabilization solution containing 0.1% Triton X-100 and 0.1% sodium citrate in 0.01 M phosphate buffered saline (PBS) for 2 min. The TUNEL reaction mixture from the in situ cell death detection kit (Roche, Basel, Switzerland) was used to treat the permeabilized tissue sections for 1 h at 37 °C in a humidified atmosphere in the dark. After the end of the TUNEL reaction, the sections were counterstained with DAPI for 5 min.

### 2.5. Immunohistochemistry

We used 0.01 M PBS in every procedure. After washing with 0.01 M PBS, the retinal sections were blocked in 10% normal donkey serum for 1 h with 0.1% Triton X-100. Subsequently, they were incubated with primary antibodies for 18 h at 4 °C. The sections were washed in PBS and incubated with a secondary antibody for 2 h at room temperature. Cell nuclei were counterstained with DAPI for 5 min. Anti-GRP78 (1:1000; Abcam, Cambridge, UK), anti-ionized calcium binding adaptor molecule 1 (IBA1) (1:500; Wako Chemical, Osaka, Japan), anti-glutamine synthase (GS) (1:1000; Chemicon, Temecula, CA, USA), anti-glial fibrillary acidic protein (GFAP; 1:1000; Chemicon) and Cy3 (1:1500; Jackson Immunoresearch, West Grove, PA, USA) or Alexa 488-conjugated antibodies (1:1000; Molecular Probes, Eugene, OR, USA) were used as secondary antibodies. Images were obtained using a Zeiss LSM 800 confocal microscope (Carl Zeiss Co. Ltd., Oberkochen, Germany).

### 2.6. Immuno-Electron Microscopy (EM) and CLEM

For immuno-EM and CLEM, we followed the protocol described by Jin et al. [25] with modifications. The retinas were dissected from the eye cup and fixed with 4% PFA for 2 h. Fixed retinas were washed with 0.1 M PB and cryoprotected with 2.3 M sucrose in 0.1 M PB for 24 h. Retinas with 2.3 M sucrose were frozen in liquid nitrogen. Semi-thin cryosections (2 μm) of the frozen retina were obtained at −100 °C using a Leica EM UC7 ultramicrotome equipped with an FC7 cryochamber (Leica). For immunohistochemistry, sections were incubated with 10% normal donkey serum for 1 h without Triton X-100 at room temperature and then double-labeled with GRP78 and GS overnight at 4 °C. To detect GRP78 in the EM, FluoroNanogold-conjugated Alexa 488 goat anti-rabbit (1:100; Nanoprobes, Yaphank, NY, USA) was used as a secondary antibody for 2 h at room temperature. The sections were counterstained with DAPI for 3 min and confocal images were obtained. After confocal microscopy, the sections were washed with 0.1 M PB and post-fixed with 2.5% glutaraldehyde and 1% osmium tetroxide for 30 min. Following post-fixation, silver enhancement was performed using an HQ silver enhancement kit (Nanoprobes) and then sections were dehydrated in graded alcohols. After the completion of all procedures, the tissues were embedded in Epon 812. Interested areas were selected in the confocal image and selected areas were cut into ultrathin sections (80–90 nm) and observed under an electron microscope (JEM 1010, Tokyo, Japan).

### 2.7. Image Analysis

Confocal images were analyzed using ZEN 2.3 software (Carl Zeiss). The immunohistochemistry intensity was measured by a histogram function and co-localization ratio automatically calculated by ZEN’s co-localization function.

### 2.8. Statistical Analysis

Quantified intensity values were statistically analyzed using Prism 8.0 software (GraphPad, San Diego, CA, USA). A one-way ANOVA was used to determine statistical significance with a *p*-value < 0.05.

## 3. Results

### 3.1. GRP78 Expression in a Normal Retina

First, we determined the GRP78 expression pattern and its localization in the normal retina at cellular and subcellular levels using immunohistochemistry and immuno-EM methods. In the normal retina, a strong immunoreactivity of GRP78 was observed in cell bodies in the INL and GCL and in the inner and outer segments (IS/OS) of photoreceptors and the outer plexiform layer (OPL) while weak punctate labeling was observed in the ONL and inner plexiform layer (IPL) (Figure 1A). To examine its localization at the subcellular level, we performed immuno-EM using a 1.4 nm gold-conjugated secondary antibody. In the ONL, gold particles were rarely detected in the cell bodies of the photoreceptor while they were mainly localized in thin cytoplasmic components within the intercellular spaces (Figure 1B), which might be the process of Müller glial cells, which are the main glial cells in the retina. In the INL and GCL, gold particles were distributed through the nuclear membrane (Figure 1C,D) and membranous structures near the nucleus (Figure 1D), which might be the ER.

### 3.2. Changes in GRP78 Expression in Blue LED-Induced RD

We subsequently examined the GRP78 expression in RD retinas at 24 h and 72 h after blue LED exposure (Figure 2) when photoreceptor degeneration started and peaked, respectively [22,23,24]. GRP78 immunoreactivity was increased in the ONL and IPL after LED exposure in a time-dependent manner (yellow rectangles in Figure 2B,F,J). In the RD retina 72 h after blue LED exposure, many GRP78-labeled lumps were observed in the ONL (yellow circles in Figure 2J). Considering that GRP78 is mainly localized in a cellular component of the intercellular spaces between photoreceptors in the normal retina (Figure 1B), which might be Müller cell processes, its cellular identity appeared to be Müller cells not photoreceptors. To confirm this point, we performed a double-label immunofluorescence experiment with anti-GRP78 and anti-GS, a representative Müller cell marker. In the normal retina, GS immunoreactivity was found in Müller cells whose bodies were situated in the middle of the INL and processes in the ONL, OPL, IPL and GCL from cell bodies to the outer and inner limiting membranes with a variety of lines in size (Figure 2C). In the RD retinas, GS immunoreactivity was remarkably increased in the ONL and thus it appeared as circles and outwardly extending thick irregular lines (Figure 2G,K,M), which suggested that Müller cell processes became hypertrophied and occupied the ONL where photoreceptors degenerated. In merged images (Figure 2D,H,L), the co-localization of GRP78 with GS in the ONL and INL was observed in approximately 23% and 41% of the normal retina, respectively, and significantly increased to 42% and 66%, respectively, in RD at 72 h after blue LED exposure (*p* < 0.05, *n* = 7 in Figure 2M,N).

However, GRP78-labeled lumps in the ONL in RD at 72 h after blue LED exposure did not show GS immunoreactivity (Figure 2L, yellow circles) suggesting that they belonged to different cellular profiles. As microglial cells occurred in similar locations in this RD model with the same morphology [22,23,24], we tested whether GRP78 was expressed in microglial cells in RD retinas by double-labeling with anti-GRP78 and anti-IBA1, a microglial cell marker. In the normal retina, IBA1-labeled microglial cells were mainly observed in the IPL and OPL (red in Figure 3A). However, in RD induced by blue LED exposure, IBA1-labeled microglial cells were mainly found in the ONL and inner and outer segment layers of the photoreceptors (red in Figure 3B,C). These microglial cells showed weak GRP78 immunoreactivity in their cell bodies and processes in normal retinas (Figure 3A) while it was increased in activated microglial cells, which migrated into the ONL where photoreceptors degenerated followed by the time course of RD (Figure 3B–C). In addition, GRP78-labeled lumps in the ONL observed in RD at 72 h after blue LED exposure were co-labeled with IBA1 (Figure 3C). These results indicated that resting and activated microglial cells expressed GRP78 in the normal and RD retinas and GRP78 expression was increased as microglial cells were activated in RD.

### 3.3. Subcellular Localization of GRP78 in Glial Cells of the Retina

GRP78 is known to localize various subcellular components including the cell membrane, nuclear membrane and cytosol [26] according to cell type. To elucidate the functional role of GRP78 in two retinal glial cells, Müller cells and microglia, we employed CLEM, an advanced EM technique combined with immunohistochemistry to determine the subcellular localization (Figure 4). As it is difficult to delineate the Müller cell processes in the INL and IPL without markers, we performed double-labeling with an anti-GS/Cy3-conjugated secondary antibody and an anti-GRP78/Alexa488-FluoroNanogold-conjugated secondary antibody in semi-thin cryosections of the RD retina. Under confocal microscopy, GS/Cy3-labeled Müller cell processes that were apparently co-labeled with GRP78/Alexa 488-FluoroNanogold were identified and selected for EM observation.

In the confocal (Figure 4A) and its correlated EM (Figure 4B) images taken from the ONL in a RD retina, two types of retinal glial cells were identified. One type was GRP78 single-labeled microglial cells without GS immunoreactivity (red asterisk in Figure 4A), which showed typical microglial cell morphology characterized by irregular nuclei with a characteristic peripheral heterochromatin and heterochromatin net (red asterisk in Figure 4B). The other was GRP78/GS double-labeled Müller glial cells (orange asterisk in Figure 4A), which had irregular nuclei in shape that were apparently distinguished from neighboring photoreceptors and microglial cells (orange asterisk in Figure 4B). In higher magnification EM views of the microglial cells (Figure 4C) and Müller cells (Figure 4D) in Figure 4B, the immuno-gold particles of GRP78 were localized to the nuclear membrane, membranous structures in the perinuclear region (orange arrows) and ER (red arrows) in the perinuclear region.

In the INL, GRP78/GS double-labeled Müller cell processes were frequently observed by confocal microscopy (Figure 4E). In correlative EM (Figure 4F,G), the immuno-gold particles of GRP78 were found in an elongated membranous structure suggesting that GRP78 was expressed in newly formed ER in hypertrophied Müller glial cells in the RD retina.

## 4. Discussion

We determined the cellular and subcellular localization of GRP78 in normal and blue LED-induced RD retinas. GRP78 is expressed in various types of retinal neurons and glial cells. In RD, GRP78 was not increased in the damaged photoreceptors, resulting in photoreceptor apoptosis while it was increased in Müller and microglial cells together with the development of a new ER.

UPR related to ER stress is expected to be a promising target for the treatment of various degenerative diseases because it is involved in cell survival and death and thus controls cell fate [27,28,29]. Among all UPR products, GRP78 belongs to the heat shock protein 70 family (HSP70), which deals with damaged or misfolded proteins to protect cells [17,18]. In previous studies [13,19,20,21], the protective role of GRP78 in response to various retinal injuries was proposed. For instance, that role against ganglion cell injury has been relatively well described [21] while GRP78′s role in photoreceptor degeneration remains unclear. Moreover, it is not clear whether photoreceptors in the normal retina express GRP78. Previous studies have shown GRP78 expression in the normal 661 W cell line, which was derived from mouse cone cells [13] while other studies have shown that GRP78 is rarely expressed in the ONL in the retinas [19] of normal and P23H rats [30] and ER stress-activated indicator (ERAI) transgenic mice [20]. We therefore wanted to describe the exact cellular and subcellular localization of GRP78 in photoreceptors.

In this study, we confirmed GRP78 expression in photoreceptors in the normal retina, albeit at low levels. At the subcellular level, it was localized to membranous structures. This expression and localization pattern was observed in RD photoreceptors but did not change (Figure 1 and Appendix A). These results appeared to be inconsistent with previous studies showing that GRP78 expression was increased in the photoreceptor cells in RD suggesting a protective role in RD by the inhibition of photoreceptor apoptosis [13,19]. First, this discrepancy might be caused by in vitro RD conditions [13,19] in which only a 661 W cell, a transformed cell line derived from the cone cell, exists and it cannot interact with the retinal pigment epithelium and/or rod and/or Müller glial cells. In addition, the finding of increased GRP78 in the ONL being due to photoreceptors might be falsely attributed [13,19]. Actually, increased GRP78 expression occurred in Müller cell processes that occupied a narrow interphotoreceptor space (Figure 2 and Figure 4), not in the photoreceptors in RD (Appendix A) as demonstrated in this study. These findings suggested that GRP78 was not directly involved in photoreceptor cell death. Instead of GRP78, apoptotic UPR pathways might be activated during photoreceptor apoptosis in RD. CHOP may be a strong candidate for cell death and is already known to be expressed in photoreceptors and plays an essential role in photoreceptor apoptosis via ER stress in RD [31,32].

Glial cells play an important role in various types of brain injury. In particular, two glial cells, microglia and astrocytes, are involved in neuroinflammation. Each type is divided into two subpopulations (M1/M2 and A1/A2) depending on its function and with an opposing role: pro-inflammatory vs. anti-inflammatory [33,34], which eventually results in neurodegeneration or neuroprotection. In neuroinflammation related to various brain pathologies, ER stress contributes to determining the phenotypes of microglia [35,36] and astrocytes [37,38]. It has been reported that GRP78 induces phagocytic function and cytokine production in microglial cells [39] and activates astrocytes to promote neuroprotective cytokines [40]. Moreover, it protects astrocytes themselves under stress conditions [41].

In this study, we demonstrated that GRP78 expression was increased in two types of retinal glial cells, Müller cells and microglial cells, in a blue LED-induced RD model. These results were consistent with the increase in GRP78 expression in glial cells in response to several types of brain and spinal cord injuries [25,42,43,44]. Müller cells are the main glial cells in the retina, corresponding to astrocytes in the brain and spinal cord. As GRP78 could activate astrocytes to promote neuroprotective cytokines [40], increased GRP78 in Müller cells might have had a neuroprotective role in this RD model. Moreover, Müller cells are responsible for retinal metabolism through the UPR as demonstrated in diabetic retinopathy [45,46,47,48]. Considering that GRP78 belongs to the HSP70 family that deals with damaged or misfolded proteins to protect cells [17,18], GRP78 in Müller cells may be involved in retinal protection as a key component of UPR. In addition, a prominent GRP78 increase in the microglia occurred 72 h after blue LED exposure when microglial activation peaked [22,23]. This suggested that GRP78 was closely associated with microglia activation, which might phagocytose and clear degenerating photoreceptors [39]. However, the exact role of GRP78 and its mechanism of action in retinal glial cells should be further investigated in a study on GRP78 in Müller cells or microglia conditional knockout animals.

GRP78 is an ER chaperone and thus was used as a representative ER stress marker. At the subcellular level, it was localized to membranous structures including the nuclear membrane and putative ER in retinal neurons and glial cells (Figure 1 and Figure 4, Appendix A). In particular, activated Müller cell processes in blue LED-induced RD retinas showed an increased GRP78 expression with ER development (Figure 4). We confirmed this by double-labeling immunofluorescence with anti-GFAP, an activated Müller cell marker [49,50], and anti-calreticulin, a representative ER marker [51,52]. The expression of calreticulin became evident in GFAP-labeled Müller cell processes in the IPL and INL (Appendix A). Based on the fact that Müller cells are responsible for metabolic modulation in the degenerative retina [45] and that it engulfs and clears the damaged photoreceptor [53,54], increased GRP78 within newly developed ER in Müller cells might therefore be essential to resolve the overloaded metabolic needs by preventing the accumulation of misfolded proteins. In addition, activated microglial cells showed higher GRP78 levels than inactivated ones (Figure 3) and calreticulin-immunoreactive lumps similar to GRP78-immunoreactive lumps in activated microglial cells in the ONL in RD retina were found at the same time and location (Appendix A) suggesting the development of a new ER. This might be reasonable because microglia could be in an overloaded condition to produce the cytokines and phagocyte the damaged photoreceptors in RD.

Finally, we want to refer to GRP78 at the mitochondria and mitochondria-associated ER membrane. Recently, growing evidence has pointed towards GRP78 playing an important function in the mitochondria in association with co-chaperones known to be involved in calcium-mediated signaling between the ER and mitochondria, important for bioenergetics and cell survival [55]. As previous reports detected GRP78 in the mitochondria [56,57] and mitochondria-associated ER membrane [58], we tried to detect mitochondrial GRP78 localization in photoreceptors and glial cells by EM. However, immuno-gold-labeled GRP78 was rarely detected in the mitochondria either in photoreceptors or glial cells. There was also no distinguishable ER region of a strong GRP78 expression close to the mitochondria (data not shown). Nevertheless, this issue needs to be further investigated in the near future.

## 5. Conclusions

In summary, we determined the cellular and subcellular localization of GRP78 in normal and blue LED-induced RD retinas. GRP78 was expressed in various types of retinal neurons and glial cells. In RD, GRP78 was not increased in the damaged photoreceptors, resulting in photoreceptor apoptosis while it was increased in Müller and microglial cells together with the development of a new ER. These results suggested that increased GRP78 played a role in glial cell activation and neuroprotective function by modulating the UPR under stress conditions. Further studies to reveal the relationship between GRP78 and ER stress and glial cell responses in RD and its mechanism are needed, considering that the findings in this study were limited to the histology.

## Figures and Tables

**Figure 1 cells-10-00995-f001:**
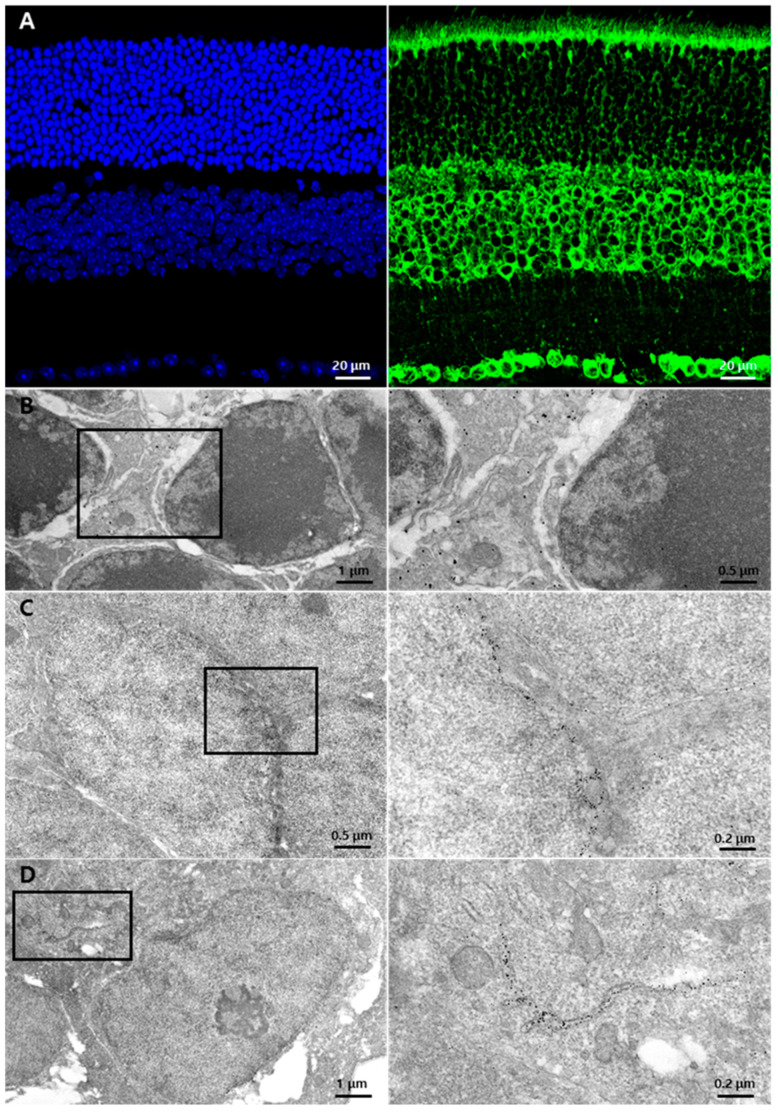
GRP78 expression in normal retinas: (**A**) Representative normal retina labeled with DAPI (blue) and GRP78 (green). GRP78 was mainly labeled in the rod and cone layers and the cell bodies of the INL and GCL. In the ONL, the photoreceptor cell bodies were weakly surrounded by GRP78. (**B**–**D**) Representative EM images of the normal retina with immuno-gold-labeled GRP78. (**B**) Normal photoreceptor cells in ONL. Immuno-gold labeled-GRP78 was detected in the interspace of the photoreceptor cell bodies but not in the photoreceptor cells. (**C**) Cell bodies in INL. Immuno-gold-labeled GRP78 was detected in the perinuclear spaces of INL cells. (**D**) Representative EM images of GCL. Immuno-gold-labeled GRP78 was detected in the perinuclear spaces and ER lumen of the GC. INL, inner nuclear layer; GCL, ganglion cell layer; ONL, outer nuclear layer; EM, electron microscopy; ER, endoplasmic reticulum; GC, ganglion cell.

**Figure 2 cells-10-00995-f002:**
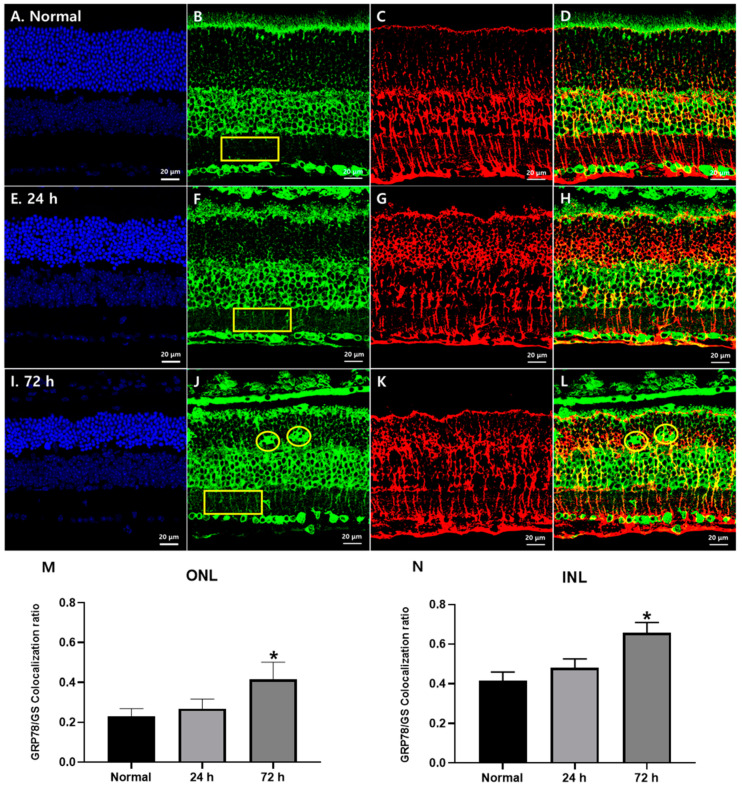
GRP78 expression in Müller glial cells in blue LED-induced RD retinas. (**A**,**E**,**I**) DAPI (blue) stained retina sections of the normal (**A**), 24 h (**E**) and 72 h (**I**) retinas after blue LED exposure. ONL thickness prominently decreased at 72 h. (**B**,**F**,**J**) GRP78 (green)-labeled sections of normal (**B**), 24 h (**F**) and 72 h (**J**) retinas after blue LED exposure. GRP78 was increased in the ONL and IPL in a time-dependent manner after LED exposure (yellow rectangles) and labeled in giant cells in the ONL (yellow circles). (**C**,**G**,**K**) GS (red)-labeled retina sections of the normal (**C**), 24 h (**G**) and 72 h (**K**) retinas after blue LED exposure. GS-labeled Müller cell processes enlarged and encircled photoreceptor cells in the ONL after injury. (**D**,**H**,**L**) Merged results of GRP78 and GS. GRP78 was increased in the Müller cell processes from IPL to ONL in a time-dependent manner while GRP78-labeled giant cells in the OPL showed no GS immunoreactivity (yellow circles). (**M**,**N**) Co-localization ratio of the GRP78/GS in the ONL (**M**) and IPL (**N**). In both the ONL and INL, the GRP78 co-localization ratio in GS-labeled Müller glial cells was significantly increased at 72 h after LED exposure (*n* = 7, *p* < 0.05 (*), ANOVA).

**Figure 3 cells-10-00995-f003:**
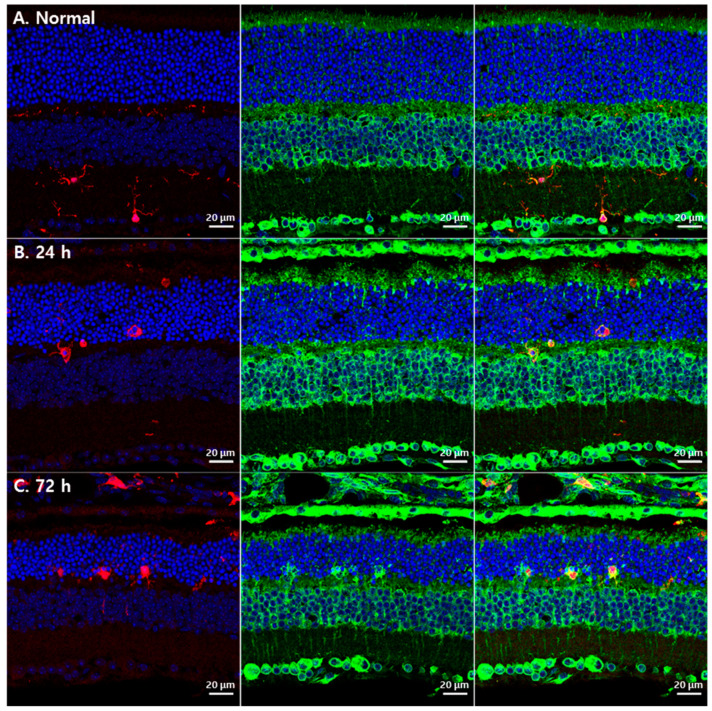
GRP78 expression in microglial cells in blue LED-induced RD retinas. (**A**) A representative confocal image of the normal retina labeled by DAPI (blue), GRP78 (green) and IBA1 (red). IBA1-labeled microglial cells were weakly labeled by GRP78 in the normal retina. (**B**). After 24 h of light exposure, microglial cells were detected in the ONL with an enlarged shape and increased GRP78 compared with the normal retina. (**C**) After 72 h of light exposure, microglial cells were markedly increased in the ONL and GRP78 was prominently increased in microglial cells compared with those after 24 h.

**Figure 4 cells-10-00995-f004:**
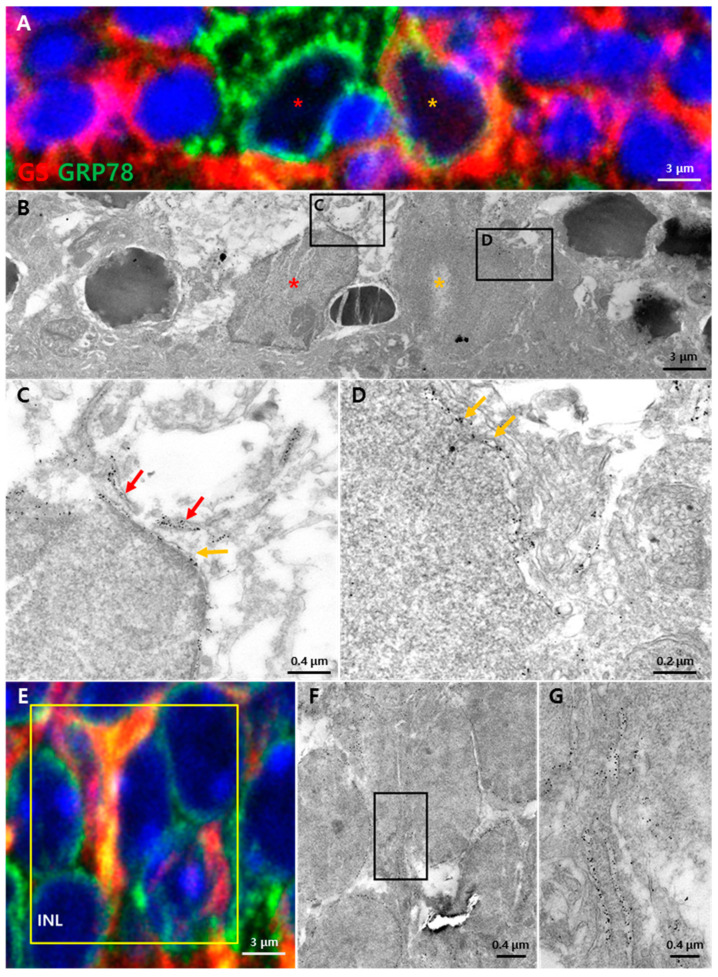
Subcellular GRP78 localization in the injured retina 72 h after light exposure. Semi-thin sections of the retina stained by GRP78 (green, gold), GS (red) and DAPI (blue). (**A**) Microglia and migrated Müller glia in the ONL. Microglia (red asterisk) were more strongly labeled by GRP78 than Müller glia without GS while Müller glia (orange asterisk) were labeled by both GRP78 and GS. (**B**) EM images of microglia (red asterisk) and migrated Müller glia (orange asterisk) in the same region as (**A**). (**C**) Immuno-gold-labeled GRP78 was detected in the perinuclear space (orange arrow) and lumen of the ER structure (red arrows) of the microglia. (**D**) Immuno-gold-labeled GRP78 was detected in the perinuclear space of the Müller glial cell (orange arrows). (**E**) Confocal images in the INL of an injured retina with Müller glia processes. (**F**) EM images matched with the yellow box in (**E**). Processes of the Müller glia in the INL labeled by GRP78. (**G**) A higher magnification of the Müller glia process in (**F**). Membranous structures containing immuno-gold-labeled GRP78 located in Müller glia processes.

## Data Availability

Data is contained within the article or Appendix A.

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
