# Peer review of "Expression of the Endoplasmic Reticulum Stress Marker GRP78 in the Normal Retina and Retinal Degeneration Induced by Blue LED Stimuli in Mice"

_cells, 2021, doi:10.3390/cells10050995_

Round 1
Reviewer 1 Report
Yong Soo Park et al show us a quite interesting work which titled Expression of the Endoplasmic Reticulum Stress Marker 2 GRP78 in the Normal Retina and Retinal Degeneration Induced by Blue LED Stimuli in Mice.
The manuscript is very well written; clear, precise, and easy to understand. I feel the overall manuscript has potential and has interesting for readers, I would be happy to suggest the editor accept the manuscript.
Anyway, I have a comment as below.
According to the publications, the ubiquitously expressed molecular chaperone GRP78 generally localizes to the ER. GRP78 is specifically induced in cells under the UPR, Cellular fractionation and protease digestion of isolated mitochondria from ER-stressed cells suggested that a significant portion of GRP78 is localized to the mitochondria and is protease-resistant. Localizations of GRP78 in ER and mitochondria were confirmed by using immunoelectron microscopy. In ER-stressed cells, GRP78 mainly localized within the mitochondria and decorated the mitochondrial membrane compartment. Submitochondrial fractionation studies indicated further that the mitochondria- reside GRP78 is mainly located in the intermembrane space, inner membrane and matrix, but is not associated with the outer membrane. Furthermore, radioactive labelling followed by subcellular fractionation showed that a significant portion of the newly synthesized GRP78 is localized to the mitochondria in cells under UPR. Those reports indicate that, at least under certain circumstances, the ER-resided chaperone GRP78 can be retargeted to mitochondria and thereby may be involved in correlating UPR signaling between these two organelles.
Does the author could discuss a little bit about GRP78 in mitochondria in the Normal Retina and Retinal Degeneration Induced by Blue LED Stimuli in Mice?
Maybe could add some perspective in the conclusion section.
Reviewer 2 Report
Comments to author
Park and coauthors, examined the protective response of GRP78 after light damage, and also determined whether if it’s located at the level of photoreceptors or not, in a light-induced animal model. The Authors (AUs) observed that the GRP78 increased in a time-dependent manner and was mainly localized in glial cells during retinal degeneration. AUs concluded that, GRP78 plays an important role in activation and neuroprotection of retinal glial cells, modulating the unfolded protein response (UPR).
- the data support the conclusions.
- the article is well written even if many typing errors and some oversights occur in the text, I recommend the authors to revise the entire article before resubmission.
Major concerns
Please carry out a thorough review of the materials and methods, some paragraphs should be described better and paragraph 2.7 “image analysis” should be separated from the statistical one.
As far as the figures are concerned, it is strongly recommended to compose them all in the same direction in order to allow the reader to immediately understand the differences.
In the discussion paragraph, line 297-304, it is advisable to better discuss the hypothesis.
In the discussion paragraph, line 318-328, please revise the entire paragraph.
minor concerns
page 1 line 21 change 24h with 24hrs and 72h with 72hrs. please correct the error throughout the article.
page 1 line 27 the phrase “The ER developed in…..lumen” is repetitive, please delete.
page 1 line 38-40 please insert wet before AMD, line 38 and delete “for the wet AMD” line 40.
page 2 line 54 change “an” with “a”.
page 2 line 66-68 the phrase “while others….[19].” Does not have sense please rephrase.
page 2 paragraph 2.2. the authors describe the light exposure technique, but if we sum all the hours it is less than 24hrs total time before sacrifice… 2hrs blue-light exposure… 1hrs dark room and then 12hrs light-dark cycle… please correct or better explain.
page 3 line 111 replace incubated with blocked
page 3 line 112 replace treated with incubated
page 3 line 136 at what does this sentence refer to “which was dehydrated in graded alcohols.
page 5 paragraph 3.2 please add arrows to the figure to better understand your results and better describe the results.
page 9 line 247 put in bold (A).
page 9 line 248 correct “iwas” with “was”.
page 9 line 277-279 please correct the entire phrase.
page 11 line 360 add GFAP in the phrase “Expression of calreticulin and GFAP….”.
Round 2
Reviewer 2 Report
Comments to the Authors
Materials and methods are now well described and appropriately used and the conclusion is well written and corroborated by the results.
However, errors and misunderstandings are still present in the manuscript, revise the figure legend using bold font for all capital letters referring the image of figure 2